# Impact-driven Context Filtering For Cross-file Code Completion

**Yanzhou Li**[1], **Shangqing Liu**[2]*, **Kangjie Chen**[1], **Tianwei Zhang**[1] **& Yang Liu**[1]
[1]Nanyang Technological University, [2]Nanjing University
{yanzhou001, kangjie001, tianwei.zhang, yangliu}@ntu.edu.sg, shangqingliu@nju.edu.cn

## Abstract

Retrieval-augmented generation (RAG) has recently demonstrated considerable potential for repository-level code completion, as it integrates cross-file knowledge with in-file preceding code to provide comprehensive contexts for generation. To better understand the contribution of the retrieved cross-file contexts, we introduce a *likelihood*-based metric to evaluate the impact of each retrieved code chunk on the completion. Our analysis reveals that, despite retrieving numerous chunks, only a small subset positively contributes to the completion, while some chunks even degrade performance. To address this issue, we leverage this metric to construct a repository-level dataset where each retrieved chunk is labeled as *positive*, *neutral*, or *negative* based on its relevance to the target completion. We then propose an adaptive retrieval context filtering framework, CODEFILTER, trained on this dataset to mitigate the harmful effects of negative retrieved contexts in code completion. Extensive evaluation on the RepoEval and CrossCodeLongEval benchmarks demonstrates that CODEFILTER consistently improves completion accuracy compared to approaches without filtering operations across various tasks. Additionally, CODEFILTER significantly reduces the length of the input prompt, enhancing computational efficiency while exhibiting strong generalizability across different models. These results underscore the potential of CODEFILTER to enhance the accuracy, efficiency, and attributability of repository-level code completion.

## 1 Introduction

Automatic code completion, particularly at the repository level, has gained significant attention due to its alignment with real-world coding scenarios. Repository-level code completion requires the model to understand the repository's domain knowledge, including cross-file contexts, to provide accurate recommendations (Zhang et al., 2023; Ding et al., 2024a). Retrieval-augmented generation (RAG) has emerged as an effective technique for integrating cross-file knowledge into the completion process. RAG-based framework first retrieves the most relevant code chunks from other files in the repository—such as user-defined APIs and inter-module dependencies—and incorporates these retrieved contexts into the prompt, which is then fed into large language models (LLMs) to enhance the completion of the current file. RAG-based methods for repository-level code completion have been extensively researched and have demonstrated substantial progress in recent years(Lu et al., 2022; Zhang et al., 2023; Liu et al.; Ding et al., 2024a).

In repository-level code completion, RAG-based methods typically rely on the preceding code snippet as a query to retrieve cross-file contexts. However, unlike natural language tasks such as question answering, where the query and relevant documents share a direct semantic relationship, the connection between the preceding code and the completed code segment is often indirect or implicit. This results in the retrieval of contexts that, despite exhibiting high semantic or token-level similarity, may not meaningfully contribute to the completion and may even degrade performance by introducing irrelevant information. Therefore, understanding the influence of each retrieved cross-file chunk is essential for optimizing the use of contextual information in code completion. Motivated by this, we

---

* Corresponding author

systematically investigate which retrieved snippets truly support the completion process and evaluate the extent to which the retrieved context is necessary for effective code generation.

To answer this question, we conduct a preliminary experiment on the popular code completion benchmark RepoEval (Zhang et al., 2023). Specifically, we define a likelihood-based metric to evaluate the impact of each cross-file chunk on the target completion. This metric measures the difference in the model's likelihood of generating the ground-truth code with and without the inclusion of a particular context (i.e., chunk) in the prompt. Applying this metric to the retrieved top-10 cross-file contexts in the RepoEval-API dataset, we find that only 15% of the retrieved chunks genuinely support the completion, while 5.6% of the chunks degrade the performance, affecting **19.81**% of the instances in the benchmark. The remaining chunks are irrelevant. These experimental results highlight that most retrieved chunks (**85**%) either do not contribute to or even hinder code completion, underscoring the need for effective filtering strategies to identify the most beneficial contexts.

In this paper, we propose an adaptive retrieval context trimming framework, CODEFILTER, to effectively select relevant retrieved contexts for repository-level code completion. The framework is trained on our constructed dataset, where each retrieved cross-file chunk is annotated with its polarity to guide the model determine whether it is beneficial for completion. Specifically, we sample 43k instances from nearly 6k diverse Python repositories, each containing consecutive lines of code for LLMs to complete. These instances are associated with over 400k cross-file context chunks, each labeled as *positive*, *neutral*, or *negative* using our proposed *likelihood*-based metric computed against the ground-truth completion. This dataset is used to train LLMs to evaluate the polarity of retrieved code chunks and retain only the positive ones as supplementary context prior to code generation. Additionally, the model is trained to adaptively determine whether the available context is sufficient for the intended completion, thereby reducing unnecessary retrieval and computation. CODEFILTER redefines the generation process with a "filtering-then-generation" paradigm, enabling the model to perform on-demand retrieval and focus only on positive retrieved contexts, which mitigates the impact of noisy or irrelevant snippets.

We conducted comprehensive experiments using different LLMs, including StarCoderBase-3B/7B (Li et al., 2023c) and CodeLlama-7B/13B (Roziere et al., 2023), on different repository-level benchmarks, including RepoEval and CrossCodeLongEval (Zhang et al., 2023; Ding et al., 2024a; Wu et al., 2024). Results show that CODEFILTER effectively filters out irrelevant retrieved content in both left-to-right and infilling code completion settings, achieving an average improvement of 3% in exact match over the baseline RAG frameworks. Moreover, CODEFILTER significantly reduces the length of cross-file contexts, shortening the original cross-file portion of the prompt by over 80% in token count. Notably, for those cases that contain negative-impact retrieved contexts, CODEFILTER successfully filters the negative contexts out, resulting in a substantial improvement of over 10% in exact match performance. Furthermore, we also establish that CODEFILTER can serve as a plug-and-play component, functioning as a retrieval context selection policy for larger models such as GPT-3.5 and improving their performance in code completion.

## 2   Related work

**Retrieval-Augmented Genration** Despite the remarkable performance of large language models (LLMs) in text and code generation, hallucination remains a significant challenge. To address this issue, retrieval-augmented generation (RAG) has emerged as a key research area, significantly enhancing generation by providing LLMs with additional accurate knowledge (Guu et al., 2020; Lewis et al., 2020), particularly in knowledge-intensive tasks such as question answering (Izacard & Grave, 2021; Ram et al., 2023; Shi et al., 2024; Borgeaud et al., 2022). Recent studies have extended RAG to programming languages by incorporating external documents or code snippets to improve code generation (Gu et al., 2016; Zhou et al., 2022; Lu et al., 2022; Zan et al., 2022). To enhance RAG efficiency and the relevance of retrieved passages, adaptive methods have been proposed to dynamically determine when additional context should be retrieved (He et al., 2021; Mallen et al., 2023; Li et al., 2023b; Jiang et al., 2023; Wang et al., 2023a; Wu et al., 2024) and how to select retrieved contexts (Wang et al., 2023b; Asai et al.; Pan et al., 2024). Moreover, RAG has been shown to be effective in addressing various code-related tasks, such as code generation (Li et al.,

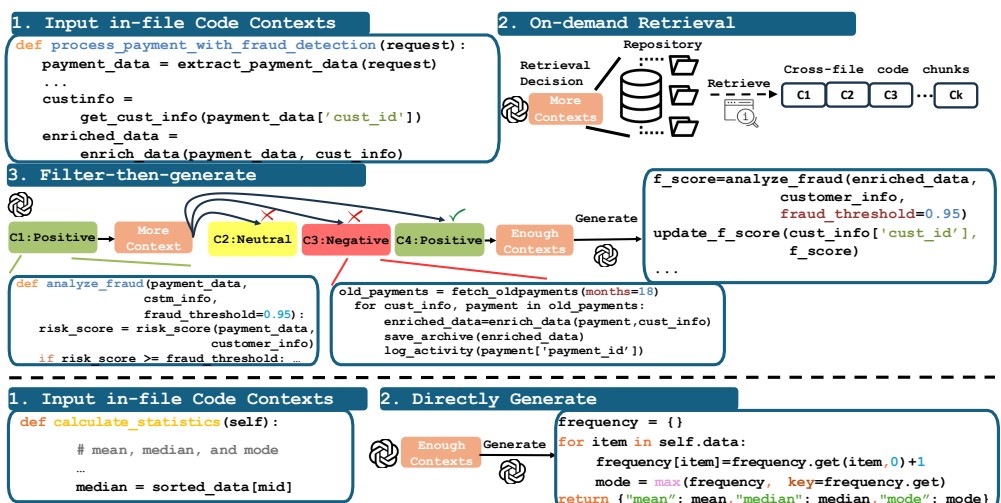

Figure 1: The overview of CODEFILTER, which initiates on-demand retrieval when the in-file context is insufficient for the intended completion; otherwise, it generates code directly. After retrieval, CODEFILTER sequentially predicts the impact of each cross-file chunk—categorized as positive, negative, or neutral—on the target completion, retaining only positive chunks. The process stops once the context is deemed sufficient, avoiding unnecessary computations.

2023a; Gou et al., 2024), summarization (Shi et al., 2022; Yu et al., 2022; Choi et al., 2023), and repair (Jin et al., 2023; Joshi et al., 2023). Our work builds on these advancements by introducing a dynamic context filtering approach tailored for code completion.

**Repository-level Code Completion** Repository-level code completion aims to enhance developer productivity by providing context-aware code suggestions. Its practical benefits and challenges in integrating comprehensive project information have garnered significant attention. Recent research has introduced benchmarks for various completion targets to evaluate the accuracy and functionality of completed code (Lu et al., 2022; Zhang et al., 2023; Ding et al., 2024a; Liu et al.; Li et al., 2024). While long-context LLMs are being explored to manage massive repository contexts (Guo et al., 2023), leveraging RAG to incorporate crucial cross-file contexts shows promise (Wu et al., 2024). Previous work primarily focused on how to format context from repositories (Cheng et al., 2024; Liu et al., 2024) and enable models to better utilize these contexts (Ding et al., 2024b; Liang et al., 2024), or on incorporating information from different modalities, such as third-party libraries (Shrivastava et al., 2023; Liao et al., 2023; Phan et al., 2024). Apart from them, our approach emphasizes understanding the impact of each code snippet and filtering retrieved contexts based on completion intent to get the model to attend to genuinely supportive information.

## 3 Repository-level Retrieval-Augmented Code Completion

### 3.1 Problem Definition

We define the components of repository-level code completion as $C_{out}, C_{in}, Y$, where $Y$ represents the target lines of code to be completed. $C_{in}$ denotes the in-file context within the target file, while $C_{out}$ refers to cross-file code from other files within the repository. To accommodate different completion scenarios, we introduce two distinct settings for $C_{in}$: (1) Infilling, where $C_{in}$ includes both the preceding and subsequent code snippets, denoted as $(C_p, C_s)$, and the model generates the missing code segment in between; and (2) Left-to-right, where $C_{in}$ consists only of the preceding code snippet $C_p$, and the model sequentially generates the subsequent code based on this context alone. The RAG-based completion framework consists of a retrieval module that uses a retriever $R$ and a generation module that leverages a generator $G$. Following previous work (Zhang et al., 2023; Ding et al., 2024a; Wu et al., 2024), we truncate cross-file contexts into chunks with a specified number of lines $C_{out} = (c_1, c_2, \ldots, c_n)$. The retriever $R$ then queries these cross-file contexts using the chunk of the preceding code of $C_{in}$ and retrieves the top-k candidate chunks with the highest similarity scores, denoted as $C_{cc} = R(C_{in}, C_{out}) = (c'_1, \ldots, c'_k)$. Given a CodeLLM as the generator $G$, the code is completed by formatting the in-file context $C_{in}$ and the retrieved context $C_{cc}$ into a single prompt, i.e., $\hat{Y} = G(C_{in}, C_{cc})$.

### 3.2 Identifying polarities of retrieved contexts

We first aim to investigate the effect of context on code completion and propose a method to identify the polarity of each code chunk as positive, neutral, or negative. Our hypothesis is as follows:

*A retrieved code chunk that contains critical information for the current completion will significantly increase the LLM's likelihood over the ground truth. Conversely, irrelevant or noisy chunks may have no effect or even decrease the likelihood.*

Based on this hypothesis, we define the contribution score $S$ of a context chunk $c_i$ to the target $Y$ as the difference in log-likelihood between a prompt containing only the in-file context $C_{in}$ and a prompt containing both $C_{in}$ and the specific code chunk $c_i$. This is expressed as:

$$S(c_i|C_{in}, Y) = \frac{L(Y \mid C_{in}, c_i) - L(Y \mid C_{in})}{L(Y \mid C_{in})}$$

Here, $L(Y \mid C)$ represents the model's log-likelihood of the target sequence $Y = (y_1, \ldots, y_T)$ given the context $C$, which is formulated as $L(Y \mid C) = \sum_{t=1}^{T} \log P(y_t \mid y_1, y_2, \ldots, y_{t-1}, C; G)$. Therefore, the polarity of $c_i$ with respect to $Y$ can be defined as:

$$P(c_i|C_{in}, Y) = \begin{cases} Positive & \text{if } S(c_i|C_{in}, Y) > T_p, \\ Negative & \text{if } S(c_i|C_{in}, Y) < T_n, \\ Neutral & \text{otherwise.} \end{cases}$$

where $T_p$ and $T_n$ represent the threshold values for determining Positive and Negative labels, respectively. In this paper, we set $T_p = 10.0\%$ and $T_n = -5.0\%$.

We evaluate the polarities of the top-10 retrieved code chunks based on the Jaccard similarity of each instance in the RepoEval dataset (Zhang et al., 2023). We compare the performance of StarCoderBase-3B in code completion using four different strategies for incorporating cross-file contexts into the prompt: (1) Full Retrieve, where all top-10 retrieved chunks are included in the prompt; (2) Positive-only, which retains only the chunks labeled as positive; (3) w/o Negative, which excludes negative chunks from the retrieved contexts; (4) w/o Neutral, which excludes chunks labeled as neutral. Results in Table (a) reveal three key findings: (1) The model with prompts containing only positive chunks outperforms the one including all candidate cross-file chunks; (2) Eliminating neutral chunks does not significantly affect the model's completion performance; (3) Removing negative chunks in the prompt improves code completion performance. These findings align with our expectations of how positive, neutral, and negative chunks impact completion, further validating the effectiveness of our likelihood-based metric by demonstrating that the model's likelihood scores can reliably indicate which retrieved contexts contribute meaningfully to the completion task.

| Strategies | Exact Match (%) |
|---|---|
| Full Retrieve | 47.27 |
| Positive-only | **49.47** |
| w/o Neutral | 47.02 |
| w/o Negative | **49.96** |

(a) The impact of different context selection strategies for completion on RepoEval-API.

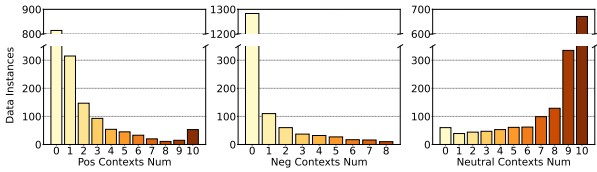

(b) Distribution of the number of positive/negative/neutral cross-file chunks for each data instance on RepoEval-API.

Furthermore, Figure (b) illustrates the distribution of positive, negative, and neutral chunks within the cross-file contexts retrieved for each instance. The x-axis represents the number of positive, negative, or neutral chunks in each instance, while the y-axis indicates the number of data instances. The data reveals that only about half of the instances contain any positive-impact chunks within their retrieved contexts, and among these, most contain only 1-2 positive chunks out of the 10 retrieved. In contrast, nearly 20% of instances include negatively impactful chunks, while the majority of retrieved chunks are neutral and irrelevant to the target completion. This distribution indicates that only a small subset of the retrieved contexts contributes meaningfully to the completion, while the rest introduce noise

or even hinder performance. To address this issue, we propose CODEFILTER, a framework that enhances efficiency by enabling on-demand retrieval and filtering out irrelevant or harmful chunks. By focusing on positive contexts, CODEFILTER improves both the performance and efficiency of code completion.

## 4 CODEFILTER

We introduce CODEFILTER, a repository-level code completion framework. This framework is illustrated in Figure 1. It is designed to (1) selectively retrieve and incrementally add code chunks and (2) filter out irrelevant chunks for improved precision and interpretability. To accomplish this, we define special signal tokens $\mathcal{T}$ in two categories. The first category comprises adaptive-retrieval tokens (<EC>, <MC>): <EC> indicates sufficient context and no further retrieval is needed, while <MC> requests additional cross-file chunks. The second category includes polarity tokens (<pos>, <neg>, <neu>), which mark a chunk's positive, negative, or neutral impact. During generation, the model learns to autonomously produce these tokens, allowing it to manage retrieval and filtering at each step.

### 4.1 Training

**Dataset Construction.** We followed the approach outlined in (Wu et al., 2024) to construct a fine-tuning dataset using the licensed repository-level dataset, Stack (Kocetkov et al.). First, we randomly sampled the target $Y$ from the raw repository data, which could be a random line, a consecutive code chunk, or an entire function body. We then retrieved the top 10 cross-file code chunks using Jaccard Similarity (Jaccard, 1912) and labeled the polarity of each chunk based on the likelihood-based metric. The detailed data construction process is provided in Appendix C.

We verbalize the training data in a fill-in-the-middle format using two strategies. The first strategy can be sequentially expressed as:

<PREFIX>[Left Context]<SUFFIX>[Right Context]<MC>[$C_1$]<pos><MC>[$C_2$]<neu>..<MC>[$C_n$]<pos><EC><MIDDLE>

where <PREFIX><SUFFIX><MIDDLE> are special tokens defined by the code LLM for the fill-in-the-middle format. Additionally, the sub-sequence <MC>[$C_1$]<pos><MC>[$C_2$]<neu>..<MC>[$C_n$]<pos><EC> represents the verbalized cross-file contexts augmented with both adaptive-retrieval tokens and polarity tokens. Moreover, the order of the candidate chunks is randomly shuffled, but the sequence includes all positive chunks, with the final chunk always labeled as positive to ensure it provides the last critical piece of information for code completion. This training data format is designed to guide the model accurately labeling the polarity of cross-file code chunks. The second format is denoted as:

<PREFIX>[Left Context]<SUFFIX>[Right Context]<MC>[$C_1$]<pos><MC>[$C_4$]<pos><MC>[$C_n$]<pos><EC><MIDDLE>[Target]

Here, the sub-sequence of cross-file chunks includes only the positive chunks. This format is designed to help the model determine whether additional information is required for completion and to complete the code based on positive cross-file code chunks. If there are no positive-labeled chunks, the cross-file chunk sequence only consists of the token <EC>, indicating that the in-file context is sufficient and no further retrieval is necessary.

**Training objectives.** Using the verbalized training dataset, we optimize the model with a standard teacher-forcing approach. This optimization is achieved by minimizing a weighted sum of the cross-entropy loss over both the signal tokens and the target tokens for code completion:

$$\mathcal{L} = -\log P_G(Y|C_{in}, C_{cc}) + \lambda(-\log P_G(\mathcal{T}|C_{in}, C_{cc}))$$

To prevent the model from memorizing irrelevant local contexts, we mask the in-file and cross-file contexts during loss calculation.

### 4.2 Inference

The inference process of CODEFILTER is designed to dynamically balance retrieval and generation for code completion.The process can be divided into four key phases:
**Analyzing In-File Context:** The model evaluates the in-file context to determine whether

additional cross-file retrieval is needed. This decision is based on generating an adaptive-retrieval token, choosing between <EC> (enough context) and <MC> (more context needed), guided by a predefined threshold applied to the softmax probability of these tokens.

**Initiating Retrieval (if needed):** If <MC> is selected, the retriever $R$ is triggered to fetch the top-K cross-file code chunks relevant to the input. These retrieved chunks are then sequentially appended to the input sequence for further evaluation.

**Chunk Evaluation and Filtering:** Each retrieved chunk is assessed by the model for relevance using polarity tokens (<pos>, <neu>, and <neg>). Positive chunks (<pos>) are retained and appended to the sequence to enrich the context. Neutral or negative chunks (<neu> or <neg>) are filtered out to avoid irrelevant or misleading information.

**Reassessing Context and Generating Code:** After adding a positive chunk, the model reevaluates whether the context is now sufficient for code generation. If <EC> is selected at this stage, the model switches to code generation using a fill-in-the-middle format. Otherwise, the process continues, iterating through the remaining retrieved chunks.

This iterative process ensures that only the most relevant cross-file contexts are used, improving the model's ability to generate accurate and efficient code completions. Moreover, we present the psudo-code of the inference process in Algorithm 1 and a generation case utilizing our framework in Appendix E for illustrating the inference process.

---

**Algorithm 1:** CODEFILTER Inference Process

---

**Input:** Generator $G$, Retriever $R$, Cross-file contexts $C_{out}$, In-file contexts $C_{in} = (C_p, C_s)$,
Adaptive-retrieval token set $\mathcal{T}_A$, Polarity token set $\mathcal{T}_P$, threshold for choosing polarity
tokens $t_p, t_n$, threshold for choosing adaptive retrieval token $t_c$

**Output:** Completed code lines $\hat{Y}$

$X \leftarrow (\text{PREFIX}, C_p, \text{SUFFIX}, C_s)$            /* Initialize input sequence */

$m \leftarrow \text{Select}(\text{Softmax}_{\mathcal{T}_A}(G(m|X)), t_c)$      /* Generate adaptive-retrieval token */

**if** $m = <EC>$ **then**
     $X \leftarrow append(X, [\text{MIDDLE}])$

**else if** $m = <MC>$ **then**
     $C_{cc} \leftarrow R(C_{in}, C_{out})$             /* Retrieve Top-K cross-file chunks */
     **foreach** *chunk* $c_i \in C_{cc}$ **do**
         $p \leftarrow \text{Select}(\text{Softmax}_{\mathcal{T}_P}(G(p|X)), t_p, t_n)$      /* Generate polarity token */
         **if** $p = <pos>$ **then**
             $X \leftarrow append(X, c_i)$
             $m \leftarrow \text{Select}(\text{Softmax}_{\mathcal{T}_A}(G(m|X)), t_c)$      /* Reassess context sufficiency */
             **if** $m = <EC>$ **then**
                $X \leftarrow append(X, [\text{MIDDLE}])$
                break             /* Context is sufficient */

**return** $\hat{Y} \leftarrow G(X)$             /* Generate final completed code */

---

# 5 Experimental Setup

## 5.1 Training & Inference

**Dataset:** Following Wu et al. (2024), we sampled 6k Python repositories from Stack (Kocetkov et al.). For each instance, we retrieved 10 cross-file chunks ranked by Jaccard Similarity and labeled them with a contribution score $S$. We then filtered out low-quality data using three criteria: (1) the target file must contain at least three local import statements; (2) target lines cannot be comments or imports and must include at least six tokens; and (3) the in-file context plus the 10 cross-file chunks must be sufficiently informative for completion. After filtering, we obtained 43k instances (400k labeled chunks). We then verbalized these instances using the two strategies from Section 4.1, producing a final dataset of 130k instances. We used 95% of the data for training and 5% for validation; more details and statistics appear in Appendix C.

**Train:** We train LLMs using different variants from two model families: StarCoderBase-3B/7B (Li et al., 2023c) and CodeLlama-7B/13B (Roziere et al., 2023). The models are optimized over 2 epochs, utilizing an initial learning rate of 2e-5, 5% warm-up steps, and linear decay. Additionally, we set $\lambda = 2.0$, a batch size of 512, and a maximum sequence length of 4096. Training is conducted on 4 NVIDIA A100 GPUs, each with 80GB of memory.

Table 1: Code completion in Infilling completion setting.

| Model | RAG Strategies | Repoeval-Line | | Repoeval-API | | Repoeval-Func | | Cclongeval-Chunk | | Cclongeval-Func |
|---|---|---|---|---|---|---|---|---|---|---|
| | | EM | ES | EM | ES | UT | ES | EM | ES | ES |
| StarCoderBase-3B | No-Retrieve | 46.56 | 68.93 | 39.09 | 65.19 | 22.42 | 39.43 | 33.57 | 62.15 | 48.62 |
| | Full-Retrieve | 56.25 | 74.72 | 47.27 | 72.69 | 27.25 | 48.34 | 38.21 | 64.05 | 46.33 |
| | RepoFormer | 57.13 | 75.47 | 49.22 | 74.06 | 27.91 | 48.70 | 39.69 | 67.67 | 48.75 |
| | CODEFILTER | **60.50** | **79.07** | **50.59** | **77.28** | **29.67** | **51.35** | **41.55** | **68.63** | **52.61** |
| StarCoderBase-7B | No Retrieve | 50.50 | 71.75 | 40.71 | 66.78 | 24.18 | 43.26 | 36.93 | 64.16 | 51.11 |
| | Full-Retrieve | 58.56 | 76.86 | 48.16 | 74.62 | 29.23 | 51.77 | 43.23 | 68.31 | 46.40 |
| | RepoFormer | 59.25 | 78.06 | 49.47 | 77.00 | 31.21 | 50.43 | 44.64 | 70.40 | 45.84 |
| | CODEFILTER | **61.44** | **80.12** | **51.09** | **78.53** | **33.41** | **53.69** | **45.97** | **71.28** | **55.37** |
| CodeLlama-7B | No Retrieve | 50.69 | 72.22 | 40.34 | 65.80 | 23.74 | 43.32 | 36.17 | 64.05 | 49.23 |
| | Retrieve | 59.06 | 77.89 | 47.59 | 72.21 | 28.79 | 51.37 | 44.29 | 68.11 | 51.92 |
| | RepoFormer | 59.19 | 78.18 | 48.34 | 74.91 | **32.09** | 51.50 | 45.74 | 68.39 | 50.92 |
| | CODEFILTER | **62.56** | **81.24** | **51.53** | **77.46** | 31.65 | **53.23** | **49.53** | **73.78** | **53.45** |
| CodeLlama-13B | No Retrieve | 52.69 | 73.63 | 41.03 | 66.89 | 25.05 | 46.08 | 40.88 | 66.22 | 51.65 |
| | Full-Retrieve | 60.31 | 77.15 | 48.66 | 73.39 | 30.55 | 53.29 | 46.17 | 69.45 | 54.18 |
| | RepoFormer | 61.00 | 80.38 | 49.28 | **78.02** | 33.19 | 53.28 | 47.74 | 70.09 | 54.17 |
| | CODEFILTER | **62.94** | **81.56** | **51.84** | 77.74 | **34.29** | **56.70** | **50.22** | **73.03** | **57.69** |

**Retrieval:** In line with previous studies (Zhang et al., 2023; Ding et al., 2024a), we divide cross-file code into chunks using a window size of 10 lines and a stride size of 5 lines. The preceding 10 lines of in-file code are then used as a query to retrieve the top-10 cross-file chunks, ranked by their Jaccard similarity scores (Jaccard, 1912). Our main experiments focus on sparse retrieval, as prior research (Ding et al., 2024a) has demonstrated that dense retrieval methods do not improve completion performance. Additionally, we evaluate the performance of CODEFILTER when using a dense retriever with UniXcoder as the encoder (Guo et al., 2022), as detailed in Appendix B.

**Inference:** We use greedy decoding for code completion. For special signal tokens, the probability threshold for <MC> is set to 0.3, and <EC> is generated otherwise. For polarity tokens, we apply a threshold of 0.3 for both <pos> and <neg>, prioritizing <pos> if it meets the threshold first, and defaulting to <neu> if neither does. Detailed ablation studies on threshold settings are provided in Appendix A. Additionally, we set the maximum token length of the prompt to 4096, with 1024 tokens allocated for the in-file context and 3072 for the cross-file chunks. We utilize vLLM (Kwon et al., 2023) to accelerate the inference process.

## 5.2 Evaluation

**Datasets:** We evaluate our model on two benchmarks: RepoEval (Zhang et al., 2023), which includes line, API, and function completion tasks derived from 14 high-quality Python repositories; and CrossCodeLongEval (Wu et al., 2024), which extends the repositories from CrossCodeEval (Ding et al., 2024a) to include chunk-level and function-level code completion tasks. We consider two completion settings in our experiments: (1) Infilling, where the model completes the middle part of the code based on both the preceding and subsequent context; and (2) Left-to-right, where the model generates code sequentially using only the preceding context. We evaluate both settings on the RepoEval and CrossCodeLongEval datasets using these benchmarks.

**Model & Baselines:** We use the same CodeLlama and StarCoderBase variants employed to train CODEFILTER. We compare CODEFILTER against three baseline retrieval configurations: (1) *No Retrieve*, where the model completes code using only in-file contexts; (2) *Full Retrieve* (Zhang et al., 2023; Ding et al., 2024a), which augments in-file contexts with the top 10 cross-file code chunks; and (3) *RepoFormer* (Wu et al., 2024), which decides whether retrieval is needed before initiating it. Because our framework for filtering contexts is orthogonal to retriever-focused approaches, direct comparisons are not appropriate. Instead, as demonstrated in Appendix B, our framework can be applied on top of various retrieval methods. Moreover, Detailed implementations of these baselines are provided in Appendix D.

**Metrics:** Following (Zhang et al., 2023; Ding et al., 2024a; Wu et al., 2024), we use the execution-based metric pass rate of Unit Tests (UT) to evaluate function-level completion in the RepoEval dataset. For all other data, we use the reference-based metrics Exact Match (EM) and Edit Similarity (ES) for evaluation.

Table 2: Code completion in Left-to-right completion setting.

| Model | RAG Strategies | Repoeval-Line | | Repoeval-API | | Repoeval-Func | | Cclongeval-Chunk | | Cclongeval-Func |
|---|---|---|---|---|---|---|---|---|---|---|
| | | EM | ES | EM | ES | UT | ES | EM | ES | ES |
| StarCoderBase-3B | No Retrieve | 33.37 | 57.94 | 27.33 | 56.11 | 17.80 | 36.63 | 23.08 | 51.09 | 42.44 |
| | Full Retrieve | 48.00 | 68.44 | 38.21 | 65.37 | 23.96 | 46.39 | 33.82 | 57.37 | 43.32 |
| | RepoFormer | 47.38 | 69.67 | 38.59 | 67.32 | 25.05 | 47.40 | 34.20 | **59.38** | 44.89 |
| | CODEFILTER | **50.50** | **71.23** | **40.84** | **70.76** | **25.49** | **48.81** | **35.61** | 59.02 | **46.40** |
| StarCoderBase-7B | No Retrieve | 35.69 | 59.64 | 28.96 | 57.51 | 19.56 | 37.54 | 27.03 | 56.16 | 51.11 |
| | Full Retrieve | 48.94 | 69.05 | 39.96 | 65.97 | 25.93 | 48.11 | 39.24 | 62.40 | 46.40 |
| | RepoFormer | 48.44 | 68.09 | 38.40 | **70.22** | 25.71 | 46.16 | 38.68 | 62.27 | 45.84 |
| | CODEFILTER | **51.32** | **71.90** | **42.15** | 69.70 | **26.59** | **49.87** | **39.65** | **64.28** | **55.37** |
| CodeLlama-7B | No Retrieve | 37.25 | 61.61 | 28.52 | 57.76 | 20.00 | 40.04 | 27.80 | 55.74 | 43.04 |
| | Full Retrieve | 50.00 | 68.47 | 40.90 | 66.33 | 24.62 | 47.64 | 38.95 | 61.47 | 50.16 |
| | RepoFormer | 48.63 | 68.97 | 38.34 | 68.29 | 26.37 | 47.52 | 37.06 | 60.49 | 48.12 |
| | CODEFILTER | **51.12** | **70.63** | **41.46** | **71.04** | **27.47** | **48.33** | **39.40** | **63.05** | **51.03** |
| CodeLlama-13B | No Retrieve | 39.25 | 62.55 | 28.89 | 58.14 | 21.76 | 41.06 | 29.07 | 55.19 | 43.62 |
| | Full Retrieve | 51.81 | 71.92 | 42.28 | 69.42 | 26.59 | 49.00 | 40.95 | **65.67** | 47.82 |
| | RepoFormer | 50.06 | 69.03 | 41.59 | 69.16 | 26.25 | 48.73 | 41.10 | 65.38 | 49.96 |
| | CODEFILTER | **52.94** | **72.76** | **42.71** | **72.59** | **27.69** | **49.99** | **41.80** | 65.34 | **54.69** |

# 6 Results & Analysis

## 6.1 Main Results

We evaluate the code completion performance of CODEFILTER in two settings across different models and compare it with several baseline RAG strategies. The results, presented in Tables 1 and 2, demonstrate that incorporating retrieved cross-file chunks significantly improves performance over models that rely solely on preceding code for generation. This improvement is evident across both reference-based and execution-based evaluation metrics. When compared to full and adaptive retrieval methods, CODEFILTER consistently achieves notable enhancements across various tasks. For example, in the infilling setting, the performance of StarCoderBase-3B under CODEFILTER framework is comparable to, or even surpasses, that of the StarCoderBase-7B model using full retrieval. Furthermore, the gains achieved by CODEFILTER in this setting mirror those observed when negative chunks are removed from the prompt, as shown in Table (a) in Section 3.2. This indicates that our method effectively filters out noise in cross-file chunks, retaining only those that positively contribute to code generation.

We also evaluate code completion performance across two settings: Infilling and Left-to-right completion. The results in Tables 1 and 2 show that incorporating subsequent code snippets significantly enhances the model's completion capabilities. For line-level and chunk-level tasks, including subsequent code results in nearly a 10% improvement compared to the Left-to-right setting under the same RAG strategy. However, for function-level completion, the benefit of incorporating in-file subsequent code is less pronounced. This may be due to the function body serving as an independent module, making subsequent code (i.e., code outside the function) less relevant to the function's content. Additionally, RepoFormer underperforms compared to the full-retrieval strategy in some tasks under the Left-to-right setting, and the improvements achieved by CODEFILTER in this setting are less substantial than those observed in the Infilling setting. We hypothesize that the model's ability to assess the utility of retrieved chunks for completion depends on its understanding of the code's intention. However, with only the preceding code available, identifying the code's intention becomes more challenging, leading to a decline in performance compared to the Infilling setting.

## 6.2 Performance on instances retrieved with negative contexts

CODEFILTER is designed to filter out noisy retrieved chunks that may negatively impact the model's completion. Although CODEFILTER demonstrates improvements over baseline RAG strategies, it remains unclear how the model performs when provided with negative chunks. To address this, we use the method proposed in Section 3.2 to identify instances containing negative chunks among the top-10 retrieved cross-file code chunks in the RepoEval-API and RepoEval-Line tasks. Out of the 1,600 test instances, we identified 285 and 166 instances containing negative chunks for API and line-level completion, respectively. We then evaluate CODEFILTER on these subsets in both the Infilling and Left-to-right settings to investigate

Table 3: Code completion in data instances containing negative cross-file chunks.

| Model | RAG-strategy | Left-to-right | | | | Infilling | | | |
| | | RepoEval-Line | | RepoEval-API | | RepoEval-Line | | RepoEval-API | |
| | | EM | ES | EM | ES | EM | ES | EM | ES |
|---|---|---|---|---|---|---|---|---|---|
| Starcoderbase-7B | No Retrieve | 7.82 | 34.92 | 3.36 | 35.98 | 17.13 | 42.92 | 11.02 | 44.04 |
| | Full Retrieve | 7.23 | 37.39 | 5.46 | 39.46 | 16.39 | 42.15 | 10.63 | 47.45 |
| | CODEFILTER | **28.92** | **57.84** | **16.03** | **55.79** | **29.28** | **62.36** | **22.83** | **63.90** |
| CodeLlama-7b | No Retrieve | 7.83 | 38.99 | 3.36 | 34.47 | 17.13 | 46.36 | 10.24 | 42.44 |
| | Full Retrieve | 6.62 | 38.01 | 6.30 | 43.24 | 12.15 | 44.23 | 9.84 | 43.14 |
| | CODEFILTER | **22.89** | **55.31** | **17.64** | **54.71** | **34.25** | **65.25** | **20.47** | **63.21** |

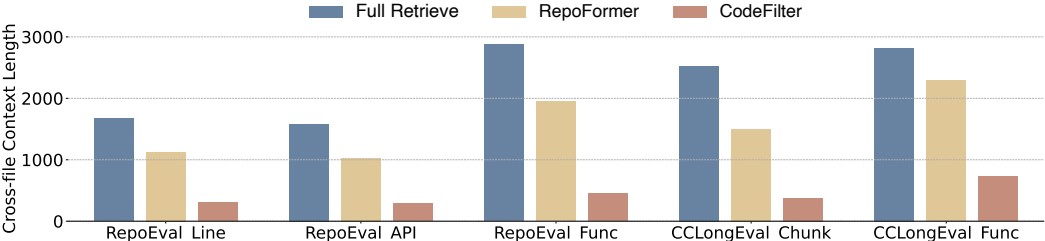

Figure 3: Cross-file Length of different benchmarks when applying different RAG Strategies.

whether the model can effectively filter out noisy information. The results, summarized in Table 3, show that full retrieval exhibits poor performance on these samples, even performing worse than directly generating code based solely on in-file context. This finding validates that these identified negative chunks directly degrade the model's completion performance. In contrast, CODEFILTER outperforms full retrieval by a significant margin across both tasks and settings, confirming that our model can effectively filter chunks based on their supportiveness for completion, thereby mitigating the impact of potential noise.

### 6.3 Length of Cross-file Contexts

In code completion, the lack of explicit information about the code's intent often results in inaccurate retrievals. To address this, a common practice is to provide the generator with up to 10 candidate chunks. However, this approach results in overly lengthy contexts, and as analyzed in Section 3.2, only a small portion of these chunks are truly relevant to the completion task. We use the length of cross-file context tokens provided to the generator as an indicator of both efficiency and attributability. Figure 3 illustrates the final cross-file context lengths under three strategies: full retrieval, RepoFormer, and CODEFILTER, evaluated across multiple benchmarks in the infilling setting. Notably, CODEFILTER refers to its StarCoderBase-3B variant, and similar context lengths were observed with other model variants of CODEFILTER after filtering. We observe that full retrieval with 10 cross-file chunks leads to excessively lengthy contexts, all exceeding 1,500 tokens. RepoFormer, which selectively determines the necessity of retrieving cross-file contexts, reduces context length by approximately 30%. Building on this, CODEFILTER further filters out irrelevant chunks, reducing context length by nearly **80**% compared to full retrieval. This reduction in context length not only improves efficiency but also increases information density, thereby enhancing the attributability of the model's completions without compromising performance.

### 6.4 CODEFILTER as filtering policy

We investigate whether the filtering decisions made by a smaller model can effectively generalize to larger models from different families and sizes. In our experiments, the StarCoderBase-3B variant of CODEFILTER is used to perform adaptive retrieval and context filtering, with the filtered prompts subsequently provided to larger models for code completion. The evaluation is conducted on the RepoEval-API task under both Infilling and Left-to-right settings. Larger models, including StarCoderBase-15B, StarCoder2-7B, CodeLlama-7B/13B, DeepSeek-16B, QWen2.5-Coder-7B, GPT-3.5-turbo, and Deepseek-R1 (Hui et al., 2024; Guo et al., 2025), are tested to assess their performance using the filtered

Table 4: Model performance on RepoEval-API when provided with cross-file contexts filtered by `CODEFILTER`-3B.

| Setting | RAG Strategy | Starcoder-15B | | Starcoder2-7B | | CodeLlama-7B | | CodeLlama-13B | | DeepSeekCoder | | QwenCoder | | GPT3.5 | | DeepSeek-R1 | |
|---|---|---|---|---|---|---|---|---|---|---|---|---|---|---|---|---|---|
| | | EM | ES | EM | ES | EM | ES | EM | ES | EM | ES | EM | ES | EM | ES | EM | ES |
| Infilling | Full Retrieve | 50.66 | 75.73 | 40.90 | 67.14 | 47.59 | 72.21 | 48.66 | 73.39 | 49.78 | 73.60 | 41.84 | 69.19 | 34.21 | 56.13 | 54.88 | **78.06** |
| | `CODEFILTER` | **51.72** | **77.08** | **42.21** | **70.33** | **49.22** | **73.34** | **49.47** | **73.78** | **51.59** | **76.05** | **45.97** | **71.41** | **36.27** | **58.75** | **55.03** | 77.49 |
| Left-to-right | Full Retrieve | 42.46 | **70.34** | 37.34 | 62.12 | 40.90 | 66.33 | 42.28 | 69.42 | 42.03 | 70.14 | 42.56 | 69.02 | 31.14 | 56.35 | 46.27 | **74.31** |
| | `CODEFILTER` | **42.90** | 69.92 | **38.84** | **63.77** | **41.65** | **69.06** | **43.46** | **70.85** | **42.96** | **71.47** | **42.63** | **69.10** | **32.02** | **57.03** | **46.32** | 73.52 |

contexts. The results, shown in Table 4, demonstrate consistent improvements in both EM and ES scores, along with significantly shorter prompt lengths when larger models generate code using the filtered contexts compared to full retrieval. These findings suggest that the filtering decisions made by the smaller model generalize well across diverse architectures, highlighting the potential of our method to serve as a plug-and-play module for enhancing both the performance and efficiency of larger models in code completion tasks.

## 7 Discussions

This work focuses on analyzing and filtering retrieved cross-file contexts for repository-level code completion. However, our current study has several limitations. First, all analyses and experiments are limited to Python, without examining whether our labeling method and `CODEFILTER` generalize to other languages. Future work will explore broader language coverage. Second, most evaluations rely on reference-based metrics. For longer targets such as code chunks or functions, execution-based metrics (e.g., unit tests) offer a more accurate quality assessment. Building such benchmarks would support deeper investigation in this domain. In addition, beyond repository-level code completion, tasks like code generation and code repair may also benefit from retrieval-augmented generation (RAG). Our methods can be extended to these tasks to examine how retrieved code influences outputs. In particular, our likelihood-based polarity metric is model-agnostic and does not depend on specific formats or domains, making it suitable for broader applications such as knowledge-intensive NLP tasks like question answering. Investigating its effectiveness across diverse natural languages is a promising direction for future work. Finally, our approach introduces some inference-time overhead. While `CODEFILTER` adds a few special-token generation steps, it reduces retrieval cost by avoiding unnecessary cross-file fetches. Compared to a full-retrieval baseline, we observe increases in inference time (+23–51%) but savings in retrieval time (22–41%), depending on the completion setting. Since retrieval and generation are typically handled by separate services, overall latency varies with system configuration. For example, with vLLM and one retrieval worker, `CODEFILTER` reduces API-level completion time from 3.4s to 1.8s; with 8 workers, the gain narrows to 0.2s. Further latency optimizations, such as parallel decoding, are left for future work.

## 8 Conclusion

In this paper, we introduced a metric to evaluate the influence of retrieved cross-file chunks on code completion and constructed a labeled dataset to categorize these chunks by their impact. We developed `CODEFILTER`, a framework that adaptively retrieves and filters relevant contexts, improving both accuracy and efficiency in code generation. Our results show that `CODEFILTER` significantly enhances performance, particularly by mitigating the effects of misleading contexts, while reducing the computational load. Additionally, the framework generalizes well across various models, demonstrating its versatility and effectiveness in repository-level code completion.

## Acknowledgments

This work was supported by the National Research Foundation, Singapore, the Cyber Security Agency under its National Cybersecurity R&D Programme (NCRP25-P04-TAICeN), the National Research Foundation Singapore and DSO National Laboratories under the AI Singapore Programme (AISG Award No: AISG2-GC-2023-008), and the National Research Foundation, Prime Minister's Office, Singapore under the Campus for Research Excellence and Technological Enterprise (CREATE) programme.

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

# A  Ablations

We ablate the impact of different thresholds for `<MC>` and `<pos>` on code completion performance and resource efficiency. The experiments are conducted using the StarCoderBase-3B model on the different tasks under the infilling setting. Figure 4 presents the results across three key metrics: Exact Match (EM), cross-file context length, and the number of generated signal tokens. Because the results across these tasks follow a similar pattern, we focus primarily on RepoEval-API in the following section.

**Thresholds vs. Exact Match**: The left column of the figure 4 shows that EM values are highly sensitive to thresholds for `<MC>` and `<pos>` tokens. For `<MC>` in Repoeval-API, EM remains stable and relatively high (50.16% to 50.59%) between thresholds 0.0 and 0.3 but drops sharply beyond this range, reaching 41.15% at a threshold of 1.0 due to disabled retrieval excluding relevant information. A similar pattern is observed for `<pos>`: a higher threshold prevents identifying positive chunks, while a lower threshold risks including noise, reducing accuracy.

**Thresholds vs. Cross-file Context Lengths**: The middle column of 4 shows that increasing thresholds for `<MC>` and `<pos>` reduces cross-file context length, improving efficiency. This effect is more pronounced for `<pos>`, with a sharp reduction as its threshold rises. At a threshold of 0.3 for both tokens, the context length is reduced to 355 tokens—less than 30% of the original—balancing minimal context length and high EM scores in Repoeval-API.

**Thresholds vs. Generated Signal Tokens**: While `CODEFILTER` improves completion performance and reduces prompt length, filtering incurs additional computational costs compared to direct generation. The number of generated signal tokens indicates this cost. As shown in the right column of figure 4, the `<pos>` threshold minimally affects this metric, while the `<MC>` threshold significantly impacts it. A lower `<MC>` threshold causes more chunks to be evaluated, increasing signal token counts, which drop sharply from 10.85 tokens at a threshold of 0.0 to 1.0 token at a threshold of 1.0 in the case of Repoeval-API.

Model performance is highly sensitive to the signal token thresholds. A low `<MC>` threshold increases resource demand without improving performance, while a high threshold enhances efficiency but reduces completion quality. Moreover, the results indicate a similar pattern across the remaining tasks. Specifically, setting the threshold for the two signal tokens to 0.2–0.3 achieves a relative balance between efficiency and performance, yielding the optimal trade-off.

# B  Generalization of `CODEFILTER` on other retrievers

We employ sparse retrieval in our main experiments, as previous work has demonstrated that sparse retrievers perform well in repository-level code completion, offering comparable results to dense retrievers. However, since our methods primarily focus on filtering the retrieved contexts, they should, in principle, be generalized to different retrieval methods, regardless of whether they are sparse or dense.

## B.1  UniXCoder

To explore the generalizability of our approach to dense retrievers, which may produce different retrieved contexts, we experimented with UniXcoder (Guo et al., 2022) as the dense retriever. We leverage UniXCoder to embed each cross-file chunk as well as the query from the in-file preceding code chunk. Then the candidate chunks are retrieved by the cosine similarity between the embedded vector of cross-file chunks and the query chunk. We maintained all indexing and query settings unchanged to ensure consistency in evaluation. Our model was then tested on the RepoEval dataset under the left-to-right and infilling settings. The results in terms of reference-based metrics are shown in Table 5.

The results show that `CODEFILTER` consistently improves performance compared to full-retrieve, which aligns with our findings from the main experiments. This suggests that `CODEFILTER` can generalize to other retrieval methods. Furthermore, when comparing the results with those from the main experiments, we observe that using UniXcoder does not

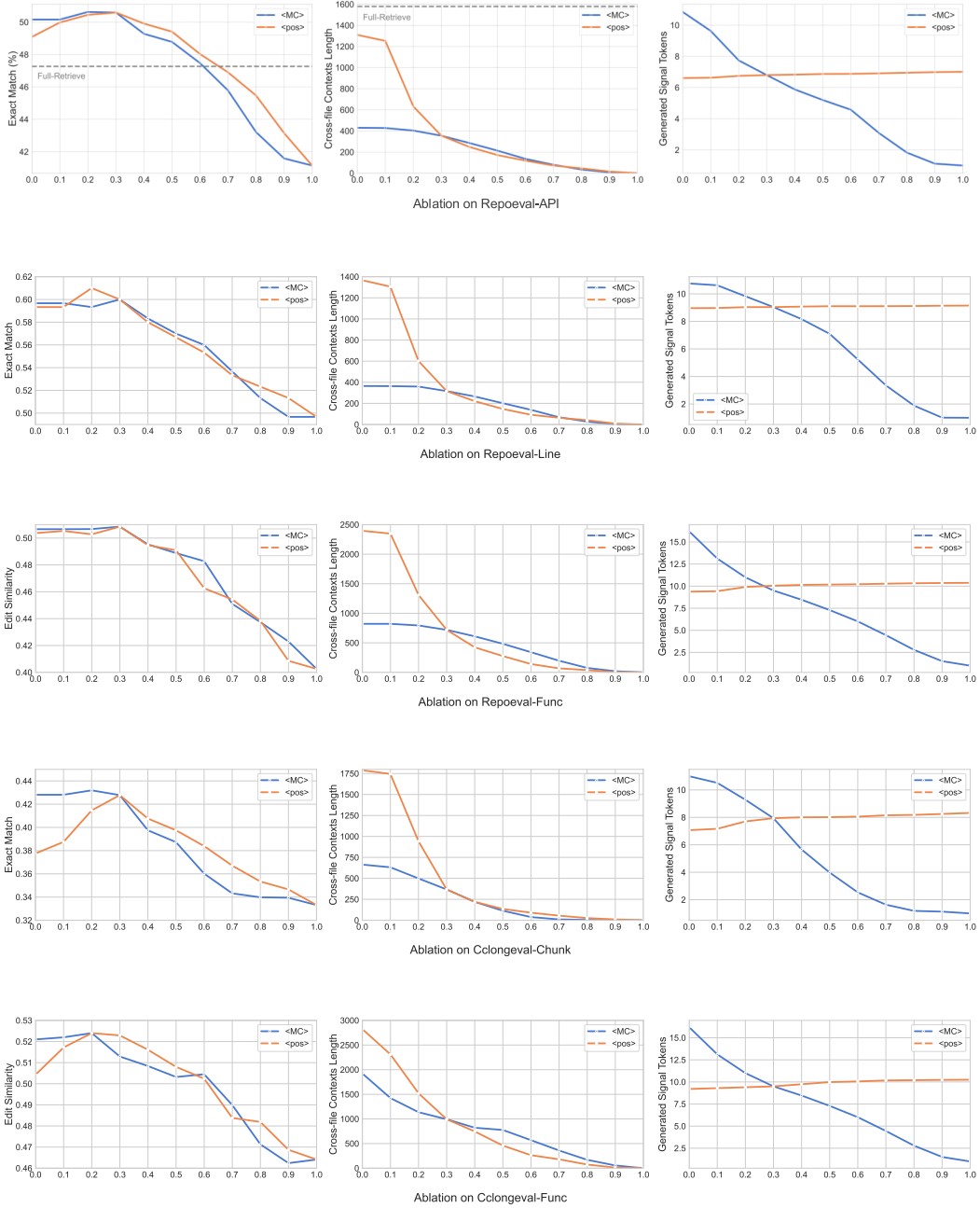

Figure 4: Ablations of threshold setting of signal tokens on extended tasks

Table 5: Code completion performance when adopting UniXcoder as a retriever.

| Completion Setting | RAG Strategy | Line | | API | | Function |
|---|---|---|---|---|---|---|
| | | EM | ES | EM | ES | ES |
| Infilling | Full Retrieve | 57.81 | 77.45 | 46.40 | 72.83 | 49.86 |
| | RepoCoder | 60.50 | 78.79 | 48.16 | 75.33 | 55.03 |
| | CODEFILTER | 60.88 | 79.53 | 49.09 | 77.62 | 53.62 |
| | CODEFILTER + RepoCoder | **61.38** | **79.80** | **50.72** | **77.80** | **55.17** |
| Left-to-right | Full Retrieve | 48.06 | 68.75 | 38.15 | 63.70 | 48.28 |
| | RepoCoder | 51.44 | 70.16 | 39.90 | 65.91 | 49.07 |
| | CODEFILTER | 51.00 | 69.52 | 40.34 | 67.48 | 48.90 |
| | CODEFILTER + RepoCoder | **52.44** | **71.64** | **40.84** | **69.17** | **50.07** |

outperform sparse retrieval and slightly reduces efficiency. This outcome is consistent with previous research (Zhang et al., 2023; Ding et al., 2024a), and we hypothesize that in repository-level code completion, the preceding code may not always convey the necessary intent for completion. As a result, semantically similar chunks retrieved by dense methods may not always contribute meaningfully to the task. While sparse retrieval also does not specifically capture the intent behind the code, its token-level similarity can help retrieve useful chunks, particularly when similar API or function names are involved. However, this approach is still suboptimal. Future research should explore methods that extract the underlying intent of the incomplete code and retrieve truly relevant chunks based on that intent.

## B.2 RepoCoder

In addition to conventional dense or sparse retrievers, some research has specifically designed retrieval frameworks tailored for repository-level code completion, aiming to retrieve more accurate and helpful cross-file contexts. A representative work in this area is RepoCoder (Zhang et al., 2023), which employs an iterative retrieval approach, incorporating the model's generated content into subsequent retrieval rounds. In our implementation, we kept all retrieval settings consistent with our main experiment and conducted two iterations of retrieval. The results for RepoCoder are also presented in the table 5, demonstrating consistent improvements over the one-time retrieval baseline. Building on this foundation, we further applied CODEFILTER to their framework to filter the final round of retrieval results. From the table, we observe that our method further enhances RepoCoder's performance, illustrating that our approach can generalize to other sophisticatedly designed retrieval frameworks.

## C  Details of dataset construction

**Data Collection:** We collected a dataset comprising 5,824 distinct projects from Stack (Kocetkov et al.), each with a minimum of 10 stars, ensuring no overlap with the RepoEval and CrossCodeEval datasets used as test sets in our main experiments. To ensure interaction with other modules within the repository, target code lines were randomly sampled from files containing at least three local import statements. These target code segments include a variety of formats to promote model generalization, ranging from single lines and chunks of 2-20 lines to full functions with fewer than 50 lines. Additionally, target lines were filtered to exclude comments and include at least six tokens. The sampled data was divided equally into two completion settings: the left-to-right setting, where only the preceding code is provided, and the infilling setting, which includes both preceding and subsequent code as in-file context.

**Data Labeling:** We then chunked the cross-file contexts and constructed queries based on the in-file context for each target $Y$. For each query, we retrieved the top 10 cross-file chunks and labeled them with a polarity—positive, neutral, or negative—using our likelihood-based metric, as detailed in Pseudo-code 2. To determine the thresholds for classifying positive and negative chunks, we experimented with various threshold settings to generate different sets of positive and negative chunks. The StarCoder-3B model's completion performance was

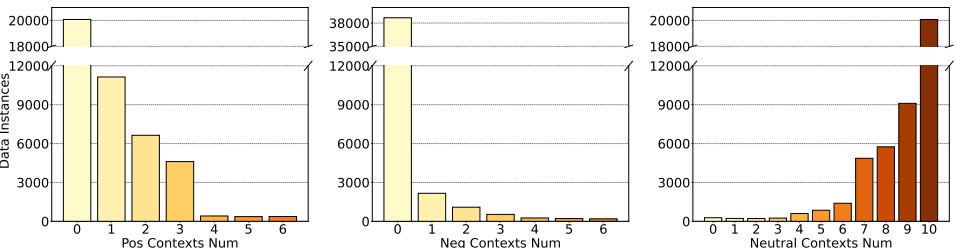

Figure 5: The statistics of chunk polarities of training data.

then evaluated on the validation set using only positive chunks or using top-10 retrieved contexts excluding negative chunks, allowing us to identify the optimal threshold. The adaptive retrieval token in our model is designed to indicate whether the current in-file and cross-file contexts provide sufficient information for code completion. To ensure this condition is met when constructing the training data, we evaluated the Edit Similarity (ES) score between the in-file context and the retrieved positive cross-file chunks. Only instances where the ES score exceeded 0.5 were included in the dataset, ensuring that the retrieved contexts contributed meaningfully to the completion task. This filtering process guarantees that the final training instances contain sufficient and relevant information. After the filtering process, we finally got **43k** instances of labelling with more than **400k** cross-file chunks. The instance-level statistics are presented in Table 6, and the chunk-level polarity distributions are shown in Figure 5. We observe that positive chunks make up 20-30% of the retrieved chunks, while 10-20% of instances contain negative chunks. The remaining chunks are generally irrelevant to the completion task, reflecting a distribution similar to the RepoEval-API test set.

---

**Algorithm 2:** Cross-file chunks labeling

---

**Input:** Generator $G$, Retriever $R$, In-file contexts $C_{in} = (C_p, C_s)$, Cross-file code $C_{out}$, Window size $w$, stride size $s$, Polarity thresholds $T_p, T, n$

**Output:** Labeled Cross-file Chunk set $\hat{C}_{cc}$

$\hat{C}_{cc} \leftarrow \varnothing$

$Q \leftarrow C_p[-w :]$                                                `/* Query */`

$C_{out} \leftarrow$ chunklize cross-file contexts with $w$ and $s$

$C_{cc} \leftarrow R(Q, C_{out})$           `/* Retrieve the top-10 cross-file contexts */`

$L(Y|c_{in}) = \sum_{t=1}^{T} logP(y_t|y_1, .., y_{t-1}, C_{in}; G)$          `/* Compute likelihood */`

**foreach** *chunk* $c_i \in C_{cc}$ **do**

    $L(Y|C_{in}, c_i) = \sum_{t=1}^{T} logP(y_t|y_1, .., y_{t-1}, C_{in}, c_i; G)$

    $S(c_i|C_{in}, Y) = \frac{L(Y|C_{in}, c_i) - L(Y|C_{in})}{L(Y|C_{in})}$

    $Polarity(c_i|C_{in}, Y) \leftarrow (S(c_i|C_{in}, Y), T_p, T_n)$         `/* Polarity of c_i */`

    $\hat{C}_{cc} \leftarrow Append(c_i, Polarity(c_i|C_{in}, Y))$

**return** $\hat{C}_{cc}$

---

# D  Implementation details

Since our methods primarily focus on how the generator utilizes the retrieved contexts, we maintain the same setup for the retrieval process across all experiments. The details of truncation, query formation, and the retrieval procedure have been introduced in previous sections. In this section, we will outline the basic prompt structure used for code generation, followed by the implementation details of both the baseline models and our proposed methods.

Table 6: Training data statistics.

| Completion Setting | Line | | Chunk | | Function | | Total |
|---|---|---|---|---|---|---|---|
| | Left-to-right | Infilling | Left-to-right | Infilling | Left-to-right | Infilling | |
| Instances | 6982 | 7056 | 7850 | 7746 | 7415 | 6561 | 43610 |

**Prompt Format:** We adopt the same prompt format for both the baselines and our proposed methods to ensure consistency. Since our experiments are conducted in both the left-to-right and infilling settings, we standardize the prompt format using the fill-in-the-middle structure for fair comparison. Additionally, following the approach in (Zhang et al., 2023; Wu et al., 2024), we represent the cross-file contexts using both natural language descriptions and the token # to denote code snippets. To provide structural information, the original file path of each code chunk is also included. A typical chunk is verbalized as:

```
#Here are relevant code fragments from other files of the repo:
# -----
# the below code fragment can be found in:
# huggingface_diffusers/examples/dreambooth/train_dreambooth_lora.py
# -----
#    pipeline = DiffusionPipeline.from_pretrained(
#        args.pretrained_model_name_or_path, revision=args.revision,
#        torch_dtype=weight_dtype
#    )
#    pipeline.scheduler = ...
#    ...
```

**Baseline Implementation:** To ensure consistency across different baselines, we formatted prompts for the no-retrieval and full-retrieval settings based on the same retrieval setup and prompt structure. In the full-retrieval baseline, cross-file contexts are ordered by their Jaccard similarity score. For the adaptive retrieval baseline i.e., RepoFormer, due to the absence of an open-source dataset as well as its model, we trained RepoFormer on the same dataset used for CODEFILTER to ensure a fair comparison. RepoFormer is designed to determine whether retrieval is necessary for code completion. To achieve this, we adapted our dataset, labeling instances with no positive cross-file samples as *no retrieval needed*, while those with relevant cross-file contexts were labeled as *retrieval needed*. This resulted in 20,076 instances labeled as the no retrieval needed and 23,534 instances labeled as the retrieval needed. Additionally, we introduced the special token <MC> to indicate whether retrieval is required in the prompt for model training. RepoFormer was trained using the same data construction process and hyperparameter settings as CODEFILTER, ensuring that the comparison between the two models remains fair and consistent.

## E   Case Study

In this section, we present a case study to illustrate how our model determines the polarity of each chunk and utilizes filtering to facilitate correct code generation. In the example depicted below, the <prefix> contains a portion of the preceding code, while the <suffix> section shows the model initiating the retrieving process by first outputting <MC>. Subsequently, the model sequentially evaluates each chunk, outputting the corresponding polarity token at the end of each chunk (denoted by a special end-of-chunk token).

When a chunk is identified with <pos>, the model further evaluates the sufficiency of the context. Once the context is deemed sufficient, the model outputs <EC> and directly proceeds with code generation.

In this case study, we compare the generation results of the baseline with full-retrieval and our framework. Notably, our framework's output exactly matches the ground truth, whereas the baseline generates incorrect results. We attribute the baseline's failure to being misled by the first negative chunk, which involves interval computations and symbolic

mappings. These suggest testing WildFunction in a more complex applied context, leading to incorrect guidance.

In contrast, the two chunks identified as ¡pos¿ provide critical support for the correct completion. One chunk shows how objects like Function are instantiated, directly informing the appropriate completion for WildFunction. Another chunk clearly explains the purpose and usage of WildFunction through its docstring, further reinforcing the correct completion.

This demonstrates how our framework effectively filters and prioritizes relevant contexts, enabling precise code generation. <PREFIX>

```
1  ...
2
3  def test_sympy__core__function__Application():
4      from sympy.core.function import Application
5      assert _test_args(Application(1, 2, 3))
6
7
8  def test_sympy__core__function__AppliedUndef():
9      from sympy.core.function import AppliedUndef
10     assert _test_args(AppliedUndef(1, 2, 3))
11
12
13 def test_sympy__core__function__Derivative():
14     from sympy.core.function import Derivative
15     assert _test_args(Derivative(2, x, y, 3))
16
17
18 @SKIP("abstract class")
19 def test_sympy__core__function__Function():
20     pass
21
22
23 def test_sympy__core__function__Lambda():
24     assert _test_args(Lambda((x, y), x + y + z))
25
26
27 def test_sympy__core__function__Subs():
28     from sympy.core.function import Subs
29     assert _test_args(Subs(x + y, x, 2))
30
31
32 def test_sympy__core__function__WildFunction():
```

<SUFFIX>

```
1  <mc>
2  # Here are some relevant code fragments from other files of the repo:
3  # -----
4  # The below code fragment can be found in Shardul555/sympy/sympy/solvers/
       ode/tests/test_single.py
5  _ode_solver_test(
       _get_examples_ode_sol_nth_linear_undetermined_coefficients)
6
7
8  def test_nth_order_reducible():
9      from sympy.solvers.ode.ode import _nth_order_reducible_match
10
11     F = lambda eq: _nth_order_reducible_match(eq, f(x))
12     D = Derivative
13     assert F(D(y*f(x), x, y) + D(f(x), x)) is None
14     assert F(D(y*f(y), y, y) + D(f(y), y)) is None
15     <End_of_Chunk> <neu>
16
17 ###################################################
```

```
18 ####################################################
19 # -----
20 # The below code fragment can be found in Shardul555/sympy/sympy/solvers/
        ode/tests/test_single.py
21 @_add_example_keys
22 def _get_examples_ode_sol_almost_linear():
23     from sympy import Ei
24     A = Symbol('A', positive=True)
25     f = Function('f')
26     d = f(x).diff(x)
27
28     return {
29     <End_of_Chunk> <pos> <mc>
30 ####################################################
31 ####################################################
32 # -----
33 # The below code fragment can be found in Shardul555/sympy/sympy/sets/
        tests/test_sets.py
34 x = Symbol('x', real=True)
35 assert Interval(0, x).free_symbols == {x}
36
37
38 def test_image_interval():
39     from sympy.core.numbers import Rational
40     x = Symbol('x', real=True)
41     a = Symbol('a', real=True)
42     assert imageset(x, 2*x, Interval(-2, 1)) == Interval(-4, 2)
43     assert imageset(x, 2*x, Interval(-2, 1, True, False)) == \
44     <End_of_Chunk> <neg>
45 ####################################################
46 ####################################################
47 # -----
48 # The below code fragment can be found in Shardul555/sympy/sympy/solvers/
        ode/tests/test_ode.py
49 def test_issue_4785():
50     from sympy.abc import A
51     eq = x + A*(x + diff(f(x), x) + f(x)) + diff
52     (f(x), x) + f(x) + 2
53     assert classify_ode(eq, f(x)) == ('1st_linear', 'almost_linear',
54         '1st_power_series', 'lie_group',
55         'nth_linear_constant_coeff_undetermined_coefficients',
56         'nth_linear_constant_coeff_variation_of_parameters',
57         '1st_linear_Integral', 'almost_linear_Integral',
58         'nth_linear_constant_coeff_variation_of_parameters_Integral')
59     # issue 4864
60     <End_of_Chunk> <neu>
61 ####################################################
62 ####################################################
63 # -----
64 # The below code fragment can be found in sympy/sympy/core/function.py
65 class WildFunction(Function, AtomicExpr):  # type: ignore
66     """
67     A WildFunction function matches any function (with its arguments).
68
69     Examples
70     ========
71
72     >>> from sympy import WildFunction, Function, cos
73     >>> from sympy.abc import x, y
74     >>> F = WildFunction('F')
75     >>> f = Function('f')
76     """
77     <End_of_Chunk> <pos> <ec>
```

**CODEFILTER Completion**

```
1  def test_sympy__core__function__WildFunction():
2      from sympy.core.function import WildFunction
3      assert _test_args(WildFunction('f'))
```

**Baseline Completion**

```
1  def test_sympy__core__function__WildFunction():
2      from sympy.core.function import WildFunction
3      from sympy import Symbol, imageset, Interval
4      x = Symbol('x', real=True)
5      f = WildFunction('f')
6      assert imageset(x, f(x), Interval(-2, 2)) == imageset(x, f(x),
           Interval(-2, 2))
```

