# OpenReview forum: "Impact-driven Context Filtering For Cross-file Code Completion"
_colmweb.org/COLM/2025/Conference — COLM 2025_

### Official Review · Reviewer_6TNq · 2025-05-01

**Rating:** 7
**Confidence:** 4
**Ethics Flag:** 1

**Summary:**

In repository-level code completion, a common approach is to retrieve relevant code chunks using retrieval-augmented generation (RAG) and include them in the prompt. However, not all retrieved code chunks help completion — some are neutral or even harmful. And including too many code chunks can exceed the context limits of large language models (LLMs). This paper tackles these challenges.

- It introduces a likelihood-based metric to label retrieved code chunks as positive, neutral, or negative based on their impact on code completion.
- A labeled dataset is built using the Stack repository-level dataset, enabling models to learn to predict the impact of code chunks.
- The authors propose CODEFILTER, a framework that selectively retrieves code chunks and filters out unhelpful chunks for completion.
- Experiments on RepoEval and CrossCodeLongEval show that CODEFILTER improves accuracy while reducing the prompt length by nearly 80%, demonstrating enhanced efficiency and effectiveness.

**Questions To Authors:**

- Figure 1 is not cited in the main text; referring to it could help readers better understand the CODEFILTER framework.
- Including example cases of neutral and negative code chunks would make the analysis more concrete and insightful.
- Line 210: “the process” -> “The process.”

**Reasons To Accept:**

- The motivation is clear and well-founded; improving RAG strategies is crucial for effective repository-level code completion.
- The CODEFILTER framework is elegantly designed, allowing the model to decide both when to stop retrieval and which code chunks are helpful.
- Experiments show that the likelihood-based metric is effective, and CODEFILTER not only improves completion accuracy but also significantly reduces prompt length.

**Reasons To Reject:**

- CODEFILTER may require more computational resources than baseline methods.
- The framework relies on multiple thresholds (e.g., T_p, T_n mentioned in Line 143, and inference thresholds mentioned in Line 255), raising concerns about its stability and sensitivity to these hyperparameters.

---

> ### Author Response · Authors · 2025-06-02
> **Response to Reviewer 6TNq**
>
> Many thanks for your thoughtful feedback. Following is our response regarding your concerns about our work:
>
> > **Q1**:
> CODEFILTER may require more computational resources than baseline methods.
>
> **A1**:
> CODEFILTER adds some overhead during generation because it sequentially evaluates each chunk and emits special tokens. However, by predicting context sufficiency, CODEFILTER avoids many unnecessary retrieval operations and thereby reduces retrieval cost. These savings largely offset the extra generation effort. In our evaluation setup, using a single retriever worker and vLLM for inference acceleration, CODEFILTER actually reduces per-instance processing time by approximately 1.5 seconds compared to a full-retrieve baseline, so the whole framework will not introduce significant additional computational cost.
>
> > **Q2**:
> The framework relies on multiple thresholds (e.g., T_p, T_n mentioned in Line 143, and inference thresholds mentioned in Line 255), raising concerns about its stability and sensitivity to these hyperparameters.
>
> **A2**:
> Regarding inference thresholds (Line 255), we performed a detailed ablation study in **Appendix B**, investigating their impact on completion performance and efficiency across various datasets and completion settings. While the results confirm that the thresholds indeed influence both effectiveness and efficiency, we successfully identified a unified setting balancing both aspects across multiple scenarios, demonstrating good generalizability. Given the importance of threshold sensitivity, we acknowledge this as a valuable future direction. We intend to further explore diverse factors, such as code complexity, programming language differences, and develop a dynamic, adaptive method to set thresholds optimally according to specific completion scenarios.
>
> The thresholds (T_p, T_n) mentioned at line 143 were determined empirically to achieve high recall when labeling positive and negative chunks, ensuring effective data collection for training. To identify these values, we sampled 10 % of the collected training data and searched for threshold settings based on completion performance under different prompt strategies: retaining only chunks classified as positive and removing chunks classified as negative. We required that both strategies outperform the full-retrieve baseline and that the presence or absence of neutral chunks have minimal impact on completion performance. Using these criteria on this subset, we selected optimal thresholds and then validated them on the RepoCoder and CrossCoderLongEval datasets, with consistent results (**Table (a)** in the paper delivers the results of different prompt strategies on the RepoEval benchmark according to our threshold setting). These experiments suggest our threshold settings generalize effectively across diverse Python datasets.
>
>
>
>
>
> > **Q3**:
> Including example cases of neutral and negative code chunks would make the analysis more concrete and insightful.
>
> **A3**:
> We included a representative example illustrating positive, neutral, and negative chunks along with baseline and CODEFILTER results in **Appendix G**. We agree more detailed interpretation of each chunk would be insightful, and will expand this analysis in our revised manuscript. We hope this addresses your concern.
>
>
> > **Q4**:
> > Writing issues: Figure 1 is not cited in the main text. Typo: Line 210: “the process” -> “The process.”
>
> **A4**:
> Thank you for pointing out these issues. We will revise the corresponding lines in our new manuscript for better clarity.

---

> > ### Comment · Reviewer_6TNq · 2025-06-09
> > **Acknowledgment of Rebuttal**
> >
> > Thank you to the authors for their response. I maintain my score.

---

### Official Review · Reviewer_75yn · 2025-05-13

**Rating:** 6
**Confidence:** 4
**Ethics Flag:** 1

**Summary:**

The paper introduces CodeFilter, a framework designed to improve retrieval-augmented code completion by addressing the issue of irrelevant or harmful retrieved code chunks. It presents a likelihood-based metric to evaluate the impact of retrieved code chunks and uses this metric to construct a dataset for training a filtering mechanism. CodeFilter adaptively retrieves and filters contexts, enhancing accuracy and efficiency in code generation. The framework demonstrates strong empirical results across various benchmarks and models, showing its potential to enhance repository-level code completion.

**Questions To Authors:**

- Could you please provide a clear explanation of the key differences between CodeFilter and the RepoFormer framework? Based on my understanding, RepoFormer optimizes the retrieval process itself, while CodeFilter optimizes the use of retrieved information. What I am confused about is how CodeFilter ensures the "filtering of irrelevant chunks"?
- To improve the paper's quality and adherence to academic standards, I recommend replacing any arXiv references with citations to the final, published versions of the works whenever available.

**Reasons To Accept:**

- The paper addresses a critical problem in retrieval-augmented code completion: the inclusion of irrelevant or harmful retrieved code chunks. It introduces a likelihood-based metric to quantify the impact of retrieved code chunks, providing a way to label data for training a filtering mechanism.

- The proposed CodeFilter framework demonstrates strong empirical results, improving code completion accuracy and reducing context length across various benchmarks and models. Moreover, the framework is shown to be generalizable across different LLMs and can be used as a plug-and-play component to enhance existing code completion systems.

**Reasons To Reject:**

- The paper needs to more clearly differentiate its contributions from existing work. For instance, the distinction between this work and the RepoFormer framework is not sufficiently explained, making it difficult to understand the novel aspects of the proposed approach.
- The reliance on Jaccard similarity for initial retrieval might limit the diversity and relevance of retrieved chunks, potentially overlooking semantically relevant but syntactically dissimilar code.
- The paper's evaluation, while comprehensive, could benefit from more analysis on the types of code completion scenarios where CodeFilter is most effective or where it might struggle.

---

> ### Author Response · Authors · 2025-06-02
> **Response to Reviewer 75yn (1)**
>
> Thank you for your reviewing and your suggestions. Following are our responses regarding your concern:
>
> > **Q1**:
> Provide a clear explanation of the key differences between CodeFilter and the RepoFormer framework. Explaining how CodeFilter ensures the "filtering of irrelevant chunks".
>
> **A1**:
>
> Thank you for your question. We will clarify the differences between our method and other baselines in the revised manuscript.
>
> Specifically, RepoFormer makes a single, global retrieve-or-skip decision: it predicts one flag indicating whether any retrieval is helpful and, if “yes,” appends all retrieved chunks to the prompt. RepoFormer treats the retrieval result as an indivisible block. CODEFILTER refines this pipeline by introducing two lightweight special tokens generated within the same decoding run: one signals that the existing context is already sufficient for completion, and the other—emitted for each chunk—marks whether to keep or drop it. As a result, the model can both avoid unnecessary retrieval and discard irrelevant chunks individually.
>
> Conceptually, RepoFormer optimizes whether to retrieve, whereas CODEFILTER also optimizes what to keep, achieving chunk-level adaptive retrieval that produces much shorter prompts (≥ 50 % prompt-length reduction) and stronger effectiveness (∼ 3 % higher ES) by filtering out chunks that could hurt completion.
>
> For "how CodeFILTER filters code chunks". Statistically, we collected a wide range of general Python repositories and annotated the retrieved code chunks. During training, the model learns which patterns of retrieved chunks consistently lead to a correct completion versus those that do not. Semantically, it builds an implicit sketch of the function’s intent, identifying the names, calls, and types still needed, and then tags chunks whose cues (identifier overlap, call-chain continuity, module proximity, type consistency) align with that intent as positive while marking others as irrelevant or negative. After enough examples, CodeFilter internalizes the mapping {inferred intent → usefulness cues → keep/drop}, so at inference it can filter irrelevant chunks without seeing the ground-truth code before generating the final completion.
>
> > **Q2**:
> The reliance on Jaccard similarity for initial retrieval might limit the diversity and relevance of retrieved chunks, potentially overlooking semantically relevant but syntactically dissimilar code.
>
> **A2**:
> Previous repository-level code completion research (e.g., RepoCoder, CrossCodeEval) compared sparse retrievers like Jaccard similarity with dense retrievers and found no substantial difference in final performance, indicating that dense methods are not necessarily superior in this task. For our main experiments, we therefore adopted the simpler and more efficient Jaccard similarity as the retrieval metric.
>
> Moreover, we also include additional experiments in Appendix C using a dense retriever (UniXcoder). The results align with prior findings: the dense retriever achieves comparable performance, and CODEFILTER provides similar improvements relative to baselines.
>
> One possible reason dense retrievers do not significantly outperform sparse ones is that, when using preceding lines as the retrieval query, dense retrievers select chunks semantically similar to the query itself, but these chunks may not necessarily reflect the underlying intent required for the target completion. As a result, such semantically similar chunks do not always provide the most relevant or helpful context.

---

> > ### Author Response · Authors · 2025-06-02
> > **Response to Reviewer 75yn (2)**
> >
> > > **Q3**:
> > >The paper's evaluation, while comprehensive, could benefit from more analysis on the types of code completion scenarios where CodeFilter is most effective or where it might struggle.
> >
> > **A3**:
> >
> > Thank you for this suggestion. Our evaluation already covers diverse scenarios, including API-level, function-level, and line-level tasks, both left-to-right and fill-in-the-middle configurations. Our methods show consistent improvement over always-retrieve baselines in general. Section 6.2 specifically analyzes cases with harmful chunks to demonstrate CodeFilter’s strengths. Nonetheless, there is a limitation: in other instances, the model primarily retains positive chunks to produce shorter prompts and make completions more attributable, but it can sometimes misclassify a genuinely useful chunk as irrelevant, degrading performance, especially when a function’s intent is difficult to infer.
> >
> > Besides, we also included additional analyses in Appendix C (comparing retrieved information from dense retrievers and RepoCoder) and case studies in Appendix G to illustrate how our method works in practice.
> >
> > We agree that further fine-grained analysis, such as examining performance differences by project complexity, coding style, repository structure, code topic, or programming language, would better characterize CodeFilter’s limitations. We will add more discussion and representative error cases to the revised manuscript and highlight deeper analyses as valuable future work.
> >
> > > **Q4**:
> > To improve the paper's quality and adherence to academic standards, I recommend replacing any arXiv references with citations to the final, published versions of the works whenever available.
> >
> > **A4**:
> > Thank you for the suggestion. We will update the revised manuscript to ensure all citations reference the up-to-date versions.

---

> > ### Comment · Reviewer_75yn · 2025-06-03
> >
> > I am sorry, but I still didn't get this part.
> >
> > "For "how CodeFILTER filters code chunks". Statistically, we collected a wide range of general Python repositories and annotated the retrieved code chunks. During training, the model learns which patterns of retrieved chunks consistently lead to a correct completion versus those that do not."
> >
> > Do the authors collect labels of whether a chunk leads to correct completion versus those that do not? I am still trying to understand the training procedure.
> >
> > Another follow-up question is whether a chunk is useful; the LLM needs to provide that signal (label), right? In that way, is there any inference time loss/speedup between Repoformer and CodeFILTER? If I assume, Repoformer brings in unnecessary chunks for completion, will that cause degenerated output from the LLM?
> >
> > Moreover, is there any study that shows CodeFILTER filters out **unhelpful** vs **harmful** chunks, leading to performance improvements?

---

> > > ### Author Response · Authors · 2025-06-04
> > > **Response to Reviewer 75yn**
> > >
> > > Thank you very much for your response and follow-up questions. Below, we provide point-by-point responses to your concerns：
> > >
> > >
> > > > Regarding your question: **"Do the authors collect labels of whether a chunk leads to correct completion versus those that do not? I am still trying to understand the training procedure."**
> > >
> > >
> > > A:Yes, we explicitly label each retrieved chunk based on whether it helps or harms the target completion. As introduced in **Section 3.2**, we define a *likelihood-based metric* that compares the model’s log-likelihood of the ground-truth code with and without a given chunk. Based on the resulting contribution score, we assign each chunk a positive, neutral, or negative label.
> > >
> > > Using this method, we construct a labeled training dataset covering over 400k retrieved chunks across 43k instances. The data construction process is described in **Section 5.1** and detailed in **Appendix D**.
> > >
> > > We then train CODEFILTER to predict chunk polarities and decide whether retrieval is needed using annotated signal tokens. As summarized in **Section 4.1**, the model is optimized with a standard teacher-forcing objective over these structured inputs.
> > >
> > >
> > >
> > >
> > >
> > > >Regarding your concern: **"Is there any inference time loss/speedup between Repoformer and CodeFILTER? Repoformer brings in unnecessary chunks for completion, will that cause degenerated output from the LLM?"**
> > >
> > > A:Both CODEFILTER and RepoFormer use adaptive retrieval, so there is no major difference in retrieval time or resource consumption. The main difference lies in the generation phase: CODEFILTER sequentially evaluates each chunk and generates polarity tokens, which introduces slight overhead.
> > > However, RepoFormer includes all retrieved chunks directly into the prompt, resulting in significantly longer input lengths. As shown in **Figure 3**, its average prompt length is over twice that of CODEFILTER, leading to higher token-level cost and slightly longer decoding.In our experimental setting, CODEFILTER's generation time is about 5–15% longer on different completion scenarios, but it saves on prompt length and token usage overall.
> > >
> > >
> > > Importantly, as you mentioned, including unnecessary chunks(as done in RepoFormer) can degrade performance. Since both CODEFILTER and our implemented RepoFormer are trained under the same setting and dataset, the consistent performance gains of CODEFILTER over RepoFormer in our main experiments(**Table 1 and Table 2**) confirm that CODEFILTER better filters harmful chunks, leading to more accurate completions. This also supports the conclusion that the performance gap arises from RepoFormer including irrelevant or harmful chunks in the prompt.
> > >
> > >
> > > >Regarding your concern: **Is there any study that shows CodeFILTER filters out unhelpful vs harmful chunks, leading to performance improvements?**
> > >
> > > A:Thank you for raising this question. Just to clarify: do you mean whether we have empirical evidence that CODEFILTER filters out both unhelpful (neutral) and harmful (negative-impact) chunks to improve performance?
> > >
> > > We address this specifically in **Section 6.2** and present the results in **Table 3**. In this analysis, we use our likelihood-based metric to identify test instances where the retrieved context contains harmful chunks. Table 3 shows that, in these cases, the Full-Retrieve baseline even underperforms the No-Retrieve baseline in some settings—demonstrating that retrieved chunks can indeed degrade completion quality.
> > >
> > > In contrast, CODEFILTER shows large performance improvements over Full-Retrieve on these same instances. Across different tasks and settings, CODEFILTER consistently achieves over 50%, and even up to 100% relative improvement in exact match scores. Since the only difference lies in filtering out irrelevant and harmful chunks, these gains directly support the conclusion that CODEFILTER effectively removes such detrimental contexts, leading to better suggestions.
> > >
> > >
> > > Once again, we sincerely appreciate your engagement with our work and the constructive suggestions you've provided. We hope our responses have addressed your concerns, and please feel free to reach out with any further questions or comments.

---

### Official Review · Reviewer_sXYr · 2025-05-13

**Rating:** 5
**Confidence:** 4
**Ethics Flag:** 1

**Summary:**

The paper introduces CODEFILTER, a framework aimed at enhancing repository-level code completion through the filtering of irrelevant or negative cross-file contexts in Retrieval-Augmented Generation (RAG) setups. The core contribution is a likelihood-based metric to assess the influence of each retrieved code chunk on completion tasks, classifying chunks as positive, neutral, or negative. The framework is trained to pre-emptively filter out less useful or detrimental chunks, thus improving completion accuracy and computational efficiency.

**Reasons To Accept:**

**Quality and Clarity:**

* The paper is well-structured and clearly articulated, with a comprehensive introduction that effectively outlines the motivation and background of the proposed method.
* The dataset construction process and methodological framework are clearly illustrated, providing a coherent flow that facilitates reader comprehension.

**Soundness:**

* The design of the dataset construction and training methodology is logically sound and well-justified, aligning with the stated objectives of the study.
* The paper presents a sufficient number of ablation studies and case analyses, effectively highlighting both the strengths and potential limitations of the proposed approach.

**Reasons To Reject:**

**Novelty and Significance:**

   * The primary concern lies in the methodological design, which appears to be a modest adaptation of previously established techniques, such as **Self-RAG** and **RepoFormer**. While the introduction of specialized tokens for context filtering is novel in the context of **CODEFILTER**, the resulting performance gains are marginal, particularly when evaluated with stronger base models like **CodeLlama 13B**. Moreover, the paper does not discuss its applicability to more recent and capable **Code LLMs**, which undermines the perceived impact of the proposed approach.
   * Additionally, the reliance on fixed-size code chunks for retrieval introduces potential limitations in capturing sufficient context, particularly for complex completion tasks. This limitation raises questions about the true extent of the performance improvements attributed to **CODEFILTER**, as the observed gains might be influenced by dataset bias rather than the effectiveness of the filtering mechanism itself.

**Minor Issues:**

   * In **Figure 3**, the term "RepoFilter" may be intended to reference "CODEFILTER," potentially leading to confusion.

---

> ### Author Response · Authors · 2025-06-02
> **Response to Reviewer sXYr (1)**
>
> Thank you for reviewing our paper, we greatly appreciate your feedback and suggestions. Here are our response regarding your concerns:
>
> > **Q1**:
> The methodological design, which appears to be a modest adaptation of previously established techniques, such as Self-RAG and RepoFormer
>
> **A1**:
> These methods are all about selective and adaptive retrieval, but differ in many ways:
>
> Self-RAG adds a separate critic LM that injects reflection tokens to decide, passage by passage, whether the main model should retrieve an entire document—a pattern suited to open-domain QA, where each paragraph usually carries an independent fact. In contrast, CODEFILTER lets the generator tag every retrieved code chunk with lightweight polarity tokens within the same decoding call, so no extra LM invocation is needed. Because repository-level code completion depends on multiple chunks that must work together, contextual sufficiency must be judged collectively. Self-RAG’s per-passage gating cannot capture this, which make self-RAG can't be directly applied to code completion tasks. Whereas CODEFILTER processes the whole sequence in one run and assesses sufficiency holistically.
>
>
> RepoFormer makes a single, global retrieve-or-skip decision: it predicts one flag indicating whether any retrieval is helpful and, if “yes,” appends all retrieved chunks to the prompt, treating the retrieval result as an indivisible block. CODEFILTER sharpens this pipeline by adding two lightweight special tokens generated in the same decoding run—one to signal that the existing context is already sufficient and another, emitted for each chunk, to mark it keep or drop—so the model can both avoid unnecessary retrieval and discard irrelevant chunks individually. Conceptually, RepoFormer optimises whether to retrieve, whereas CODEFILTER also optimises what to keep, achieving chunk-level adaptive retrieval that produces much shorter prompts(>=50% prompt length reduction) and stronger effectiveness(3% higher ES) by filtering out chunks that could hurt the completion
>
> > **Q2**:
> The resulting performance gains are marginal, particularly when evaluated with stronger base models like CodeLlama 13B
>
> **A2**:
> Our model improves performance by filtering retrieved noisy chunks that degrade completion. Because such instances with negative retrieved chunks account for only ≈ 10–20 % of the general test set, the overall gain on the full benchmarks is meaningful and nontrivial.
> When we isolate this difficult subset (identified in section 6.2), Improvements are much larger: relative gains over the baseline exceed 50 % across all metrics, and some even surpass 100 %. We have also added CodeLlama-13B results on this targeted subset in the table below, demonstrating similarly significant performance gains compared to the full-retrieve baseline.  These numbers show that CODEFILTER is crafted for, and delivers sizable gains on, the hard cases where unfiltered retrieval degrades performance.
>
> | Model           | RAG-strategy  | L2R Line EM (%) | L2R Line ES (%) | L2R API EM (%) | L2R API ES (%) | Infill Line EM (%) | Infill Line ES (%) | Infill API EM (%) | Infill API ES (%) |
> |-----------------|---------------|-----------------|------------------|----------------|----------------|---------------------|---------------------|--------------------|--------------------|
> | CodeLlama-13B   | No Retrieve   | 9.04            | 38.43            | 8.42           | 40.17          | 20.48               | 44.31               | 14.39              | 45.06              |
> |                 | Full Retrieve | 8.43            | 40.17            | 7.37           | 38.67          | 22.29               | 47.50               | 18.94              | 47.19              |
> |                 | CODEFILTER    | 27.11           | 59.90            | 18.25          | 53.95          | 36.75               | 67.28               | 27.72              | 67.24              |
>
> Beyond accuracy, the method also cuts prompt length by > 50 %, lowering cost and making completions more clearly attributable to the retained context across all instances.

---

> > ### Author Response · Authors · 2025-06-02
> > **Response to Reviewer sXYr (2)**
> >
> > > **Q3**:
> > The paper does not discuss its applicability to more recent and capable Code LLMs, which undermines the perceived impact of the proposed approach.
> >
> > **A3**:
> > Because of limited computational resources, we fine-tuned only two representative model families—StarCoder and CodeLlama. Beyond that, **section 6.4** tests a plug-in setting where the fine-tuned model is used solely as a context selector, and newer models handle generation. We pair the selector with several recent LLMs, including open-source QWen2.5-Coder-7B and DeepSeek-Coder-v2-Lite(16B), and commercial LLMs such as ChatGPT-3.5 and DeepSeek-R1(671B). The results in section 6.4 show that the selected chunks continue to improve their performance with much shorter prompts. These results demonstrate that our selection policy generalises well across various code LLMs. We hope these experiments confirm our method’s effectiveness with a wide range of models, including recent Code LLMs, and thus alleviate your concern.
> >
> > > **Q4**:
> > Using fixed-size code chunks may miss crucial context in complex tasks, calling into question whether CODEFILTER’s gains reflect true filtering effectiveness or merely dataset bias.
> >
> > **A4**:
> > Fixed-size code chunks are the most widely used indexing method in the retrieval-augmented code completion area in previous research, such as RepoCoder, CrossCodeEval, and RepoFormer. This method normalises chunk length, simplifies indexing, and gives each candidate comparable information content.  Because our study targets **post-retrieval selection**, not retrieval itself, we adopt this most representative fixed-size scheme for **all baselines** as well as our method during experiments, so we believe that any performance gains stem from the filtering mechanism, not the choice of indexing methods.
> > Moreover, our experiments span two representative datasets and 5 task configurations (API-level, line-level, function-level; left-to-right and fill-in-the-middle completion).  CODEFILTER shows consistent gains across every setting and two different datasets. We hope these results could validate that the performance gains are not influenced by dataset bias.
> >
> > > **Q5**:
> > Typo of legend and bars in Figure 3
> >
> > **A5**:
> > Thank you for identifying this typo. We will correct it in the revised manuscript.

---

> > > ### Comment · Reviewer_sXYr · 2025-06-07
> > >
> > > Thank you for the clarification and the additional results. I’ll stick with my current score for now.

---

> > > > ### Author Response · Authors · 2025-06-07
> > > > **Response to Reviewer sXYr**
> > > >
> > > > Thank you for your response. If you have any remaining questions, we’d be happy to provide more details or analysis. We truly value your feedback and hope to improve our work with your suggestions.

---

### Official Review · Reviewer_s8bE · 2025-05-18

**Rating:** 6
**Confidence:** 4
**Ethics Flag:** 1

**Summary:**

This paper introduces CODEFILTER, a framework to enhance repository-level code completion by filtering retrieved cross-file code chunks. It uses a likelihood-based metric, $S(c_i|C_{in}, Y)$ , to label chunks as positive, neutral, or negative during dataset creation. CODEFILTER is then trained to predict these polarities (e.g., `<pos>`, `<neg>`) and to decide if more context is needed (`<MC>`, `<EC>`) during an iterative "filter-then-generate" process .

**Reasons To Accept:**

1.  The paper tackles the critical issue of irrelevant or harmful context in RAG for code completion , which often plagues existing systems.
2.  The claims are well-supported by extensive experiments showing consistent gains across various settings and models. The efficiency gains from context reduction are also significant.
3.  The framework shows good generalization to different models.

**Reasons To Reject:**

- I think low latency is crucial for code completion scenarios. The iterative process might introduce additional time costs. I think it would be beneficial if the authors could provide some data on the total time taken for a single completion task, comparing CODEFILTER with baselines, to better understand this trade-off.
- This is perhaps more of a general thought than a weakness of the paper itself. For commercial applications, many might lean towards using API-based LLMs. The multiple steps of retrieval, model-based filtering (which itself is an LLM call per chunk in CODEFILTER's setup), and then final generation could potentially increase costs or complexity when using pay-per-call APIs. While efficiency is gained in the final generation step due to reduced tokens, an analysis of the overall "query cost" in such a scenario might be relevant for adoption.
- The paper rightly claims that reducing the input token count is a benefit, especially when models had more restrictive context windows (e.g., 4k tokens, as mentioned in the paper's setup ). With models now supporting much larger context lengths, I'm somewhat curious about the trade-off. If the time overhead of filtering is non-negligible, one might wonder if feeding more (less filtered) context to a large-context model could achieve comparable or even better results within a similar total time budget, especially if the performance gains from filtering, while consistent, are not always dramatically substantial across all tasks. I acknowledge this comparison might not be entirely straightforward, but it's a perspective worth considering as model capabilities evolve.

minor:
- In Figure 3 , the legend and bars refer to "RepoFilter". I suspect this might be a previous naming for the proposed method or a typo.

---

> ### Author Response · Authors · 2025-06-02
> **Response to Reviewer s8bE**
>
> Thank you sincerely for your valuable suggestions. Here are our responses regarding your concerns about our work:
>
> > **Q1**:
> Time costs: provide some data on the total time taken for a single completion task, comparing CODEFILTER with baselines
>
> **A1**:
> We thank the reviewer for prompting this analysis.
>
> We share your emphasis on latency.  CODEFILTER makes a deliberate trade-off: the iterative loop adds a handful of special-token generation steps, **but adaptive retrieval eliminates many costly retrieval calls**.  Under our current implementation and hyperparameters, we observe:
>
> | Phase                                  | RepoEval-Line | API-level | Function-level |
> |----------------------------------------|---------------|-----------|----------------|
> | **Generation time (+ vs. full-retrieve)** | +51 % | +44 % | +23 % |
> | **Retrieval time (– vs. full-retrieve)** | –22 % | –41 % | –36 % |
>
>
>
> Because retrieval and generation are typically deployed as independent services, the overall time overhead depends on the configuration of these two modules.  In our default evaluation setup (A100-80 GB GPU, vLLM, one retrieval worker), **CODEFILTER saves 1.5 – 1.6 s per API-level completion**.  With 8 retrieval workers the saving shrinks to ~0.2 s, while an extreme setting of 30 workers and a non-accelerated generator incurs a modest 0.8 – 0.9 s overhead.
>
> These results illustrate that hardware (GPU/CPU type), worker-count, and acceleration frameworks all influence latency, yet in typical deployments CODEFILTER *reduces or only minimally affects* end-to-end time.  We will add this discussion and the full latency table to our new manuscript.
>
>
>
> > **Q2**:
> Analysis of the overall "query cost"
>
> **A2**:
> We primarily measure query cost in terms of token usage—i.e., the number of input and output tokens incurred when calling an LLM API for code generation. Moreover, there could be some strategies to avoid excessive per‐chunk API calls. As the result, CODEFILTER is cost‐efficient in both deployment modes:
>
> 1. **Integrated API.** When the generation model is accessible, decoding with polarity tokens can be deployed in the API. Therefore, filtering occurs **within the same generation call** and no per‐chunk requests are required. Using the OpenAI pricing ratio (≈ 1 unit per input token: 4 units per output token), the total paid units per query are ≈ 55 % of the full‐retrieve baseline.
>
> 2. **External filter (Section 6.4).** With an unmodified API (e.g., GPT-4), filtering runs locally on a lightweight model such as StarCoder-7 B; its compute cost is negligible compared to API billing. The subsequent generation call uses a much shorter prompt, still reducing overall cost by > 50 %.
>
> Thus, even under pay‐per‐call billing, CODEFILTER remains more cost‐effective.
>
>
> > **Q3**:
> Trade-off between the time spent on filtering and simply feeding more raw context into larger-window models to achieve similar or better results within the same time budget.
>
> **A3**:
> As discussed in A1, the filter step adds a generation overhead but removes many retrieval calls, so end-to-end latency is essentially unchanged.  Therefore, the efficiency–effectiveness trade-off is negligible.
> Moreover, though the current model supports very long input, context is also *not* “free”: attention cost still grows with input length, and API billing remains token-based.  Crucially, extra context does not always help.  Section 3.2 shows that ≥ 80 % of retrieved lines are irrelevant, and Section 6.2 reports that unfiltered context hurts completion quality on 10–20 % of instances.  Feeding all chunks into a large-context model therefore risks both higher cost and accuracy regressions, whereas CODEFILTER keeps prompts concise and reliably improves performance.
>
> > **Q4**:
> Typo of legend and bars in Figure 3
>
> **A4**:
> We really appreciate you pointing out this typo; we will revise it in our new manuscript.

---

> > ### Comment · Reviewer_s8bE · 2025-06-08
> >
> > Thank you for the clarification. I appreciate the authors' efforts in addressing the concerns. I maintain a slightly positive evaluation of the paper.

---

### Decision · Program_Chairs · 2025-07-08

**Decision:**

Accept

**Comment:**

This paper presents analysis and methodology surrounding filtering code in RAG-for-code contexts. Specifically, many retrieved snippets have a neutral-to-negative impact on the likelihood of the correct completions. CODEFILTER is a method trained to predict the utility of retrieved snippets, which can then be used to filter snippets and determine when enough snippets have been retrieved to generate. Experiments show that CODEFILTER improves over other retrieval strategies when paired with open-source generator models like CodeLlama-13B.

This paper presents a clear, intuitive idea and demonstrates results across a range of settings. It's generally a strong piece of work.  Although the core ideas of learning this retrieval system have been explored before (RePLUG, Self-RAG, RepoFormer), I agree with the authors' response to reviewer sXYr that there is still novelty in the combination of the domain (code) and particular technical approach chosen here.

I think one main concern for the impact of the work is that more recent LLMs will not require this filtering, both due to growing context lengths and due to distractors being less damaging. However, there is some evidence in the literature that even long-context LLMs benefit from filtering, so I think this problem will not be solved immediately.